# Continuous Infusion of High Doses of Cefepime in Intensive Care Unit: Assessment of Steady-State Plasma Level and Incidence on Neurotoxicity

**DOI:** 10.3390/antibiotics12010069

**Published:** 2022-12-30

**Authors:** Vanessa Jean-Michel, Corentin Homey, Patrick Devos, Pierre-Yves Delannoy, Nicolas Boussekey, Thomas Caulier, Olivier Leroy, Hugues Georges

**Affiliations:** 1Service de Réanimation Médicale et Maladies Infectieuses, Hôpital Chatiliez, 135 rue du Président Coty, 59200 Tourcoing, France; 2University Lille, CHU Lille, Lillometrics, 59000 Lille, France

**Keywords:** cefepime, continuous infusion, intensive care unit, neurotoxicity

## Abstract

Continuous infusion (CI) with high doses of cefepime is recommended in the empirical antimicrobial regimen of critically ill patients with suspected Gram-negative sepsis. This study aimed to determine factors associated with cefepime overdosing and the incidence of cefepime-induced neurotoxicity (CIN) in these patients. We performed a retrospective study including all patients receiving cefepime treatment between January 2019 and May 2022. The plasma level of cefepime defining overdosing was over 35 mg/L. Neurotoxicity was defined according to strict criteria and correlated with concomitant steady-state concentration of cefepime. Seventy-eight courses of cefepime treatment were analyzed. The mean cefepime plasma level at steady state was 59.8 ± 29.3 mg/L, and overdosing occurred in 80% of patients. Renal failure and a daily dose > 5 g were independently associated with overdosing. CIN was present in 30% of patients. In multivariate analysis, factors associated with CIN were chronic renal failure and a cefepime plasma concentration ≥ 60 mg/L. CIN was not associated with mortality. Overdosing is frequent in patients receiving high doses of cefepime by CI. Steady-state levels are higher than targeted therapeutic pharmacokinetic/pharmacodynamic objectives. The risk of CIN is important when the plasma concentration is ≥60 mg/L.

## 1. Introduction

Cefepime is a commonly used broad-spectrum beta-lactam administered in critically ill patients with severe hospital-acquired Gram-negative sepsis [1]. Cefepime treatment can lead to neurological disorders with a large panel of clinical manifestations, ranging from myoclonus, confusion and seizures to coma [2]. Factors associated with neurotoxicity are renal dysfunction, inappropriate dosing, obesity and advanced age [2,3].

Because of its relatively narrow therapeutic–toxic window, the therapeutic drug monitoring (TDM) of cefepime is increasingly used in order to achieve an adequate pharmacokinetic/pharmacodynamic (PK/PD) index and to prevent overdosing and subsequent neurotoxicity. Higher cefepime minimal inhibitory concentration (MIC) breakpoints range from 4 mg/L for *Enterobacterales* spp. to 8 mg/L for *Pseudomonas* spp. [4]. A recent guideline suggests targeting a free plasma beta-lactam concentration between 4 and 8 times the MIC of the causative bacteria for 100% of the dosing interval (fT ≥ 4–8 × MIC = 100%) to maximize bacteriological and clinical responses in critical care patients [5]. So, when starting empirical treatment, the therapeutic plasma target of cefepime should range between 20 and 40 mg/L for most patients and reach 60 mg/L when *Pseudomonas* spp. sepsis, with a MIC value of 8 mg/L, is suspected.

Studies reporting the cefepime plasma level and threshold for neurotoxicity are scarcely available. Retrospective studies have suggested that the cefepime trough concentration (Cmin) should be maintained below 20–38 mg/L [3,6,7,8]. These studies concerned patients with the intermittent administration of cefepime. However, the continuous infusion (CI) of beta-lactams is now recommended for maximizing the blood concentration above the MIC and optimizing PK/PD targets in patients who are more likely to have less-susceptible Gram-negative infections [5]. Just one study has assessed the TDM of cefepime in patients with CI [9]. This recent single-center retrospective study determined a cefepime steady-state concentration of 63.2 mg/L as the best cut-off point between patients with and without neurotoxicity. However, this study reports measurements in non-intensive care unit (ICU) patients with a relatively low dose of cefepime (4 g/d). In our unit, cefepime is delivered exclusively by CI with a higher daily dose (6 g/d), as recently recommended for the empirical treatment of suspected severe Gram-negative sepsis [1,10]. To our knowledge, there are no data about the TDM of cefepime when high doses are delivered, most particularly in critical care patients undergoing rapid physiological changes, such as an altered fluid status or changes in serum albumin levels, resulting in high inter-individual variability in the volume of distribution and clearance.

The aims of our study were (1) to determine the mean steady-state plasma concentration of high doses of cefepime delivered by CI in critically ill patients and factors associated with overdosing and (2) to determine factors associated with cefepime-induced neurotoxicity (CIN) and to define a threshold associated with neurotoxicity.

## 2. Materials and Methods

### 2.1. Study Design and Patients

We conducted a retrospective study using prospectively collected data in a 16-bed medical and surgical adult intensive care unit (ICU) of a 450-bed general hospital. All adult patients (≥18 years old) receiving cefepime treatment between January 2019 and May 2022 were included.

### 2.2. Collected Data

The following data were recorded for each studied patient on ICU admission: age, gender, the severity of the underlying disease using the Charlson score, simplified acute physiologic score (SAPS II), underlying neurological diseases (status epilepticus, previous meningitis, encephalitis, or stroke), cirrhosis, chronic renal failure, cardiac chronic failure, diabetes and body max index [11,12].

Chronic renal failure was defined as a baseline estimated glomerular filtration rate (eGFR) <60 mL/min/1.73 m^2^ for a duration of at least 3 months.

The following data were recorded on the day of cefepime delivery: origin of sepsis, duration of ICU stay before cefepime treatment, daily dose of cefepime, Sequential Organ Failure Assessment (SOFA) score, serum albumin level and Cr/Cl [13]. The daily dose of cefepime was the dose delivered per day after the initial bolus of 2 g, before the TDM of cefepime. Infections were classified as suspected or microbiologically proven. Microbiologic evidence proving infection included positive blood cultures, endotracheal aspirates, urine, intra-abdominal fluid and bone cultures. The determination of MIC was performed for each isolated pathogen according to the EUCAST definition [4].

Steady-state plasma concentrations of cefepime were determined by high-pressure liquid chromatography (HPLC) with ultraviolet detection within 24 to 48 h following the delivery of cefepime.

The duration of ICU stay and survival status on ICU discharge were recorded.

### 2.3. Cefepime Treatment and Assessment of Overdosing and Neurotoxicity

A 2 g loading dose of cefepime administered over 30 min was delivered to all patients, immediately followed by a CI. The dose of cefepime for CI was delivered at the discretion of the treating physician according to the severity of the disease. However, it was recommended to consider renal function (RF): 2 g every 8 h for patients with an eGRF ≥ 60 mL/min and 1 g every 8 h for patients with an eGRF < 60 mL/min. RF was determined by the estimated glomerular filtration rate (eGFR) using the Chronic Kidney Disease Epidemiology Collaboration (CKD-EPI) equation [14]. The dose of cefepime was considered appropriate or not according to renal function (RF).

The cefepime plasma concentration defining overdosing was 35 mg/mL. This level was defined by the SFAR (French Society of Anaesthesia and Intensive Care Medicine, Paris, France) guidelines and was used by our laboratory [5]. Only the first TDM performed within 24 to 48 h after starting treatment was studied for each patient.

The presence or absence of CIN was determined on the day of TDM. Neurotoxicity was only determined in patients without sedation (midazolam, sufentanil or propofol). Neurotoxicity was defined by neurological symptoms of any grade documented in the patient’s medical record [15]. CIN was defined if the following two criteria were present [16]: first, the onset of CIN-defined neurological symptoms occurring at least 48 h after the commencement of cefepime therapy and with symptom resolution or improvement within 48 h of cefepime cessation or dose reduction. CIN-defined neurological symptoms were established with the occurrence of confusion, hallucinations, altered level of consciousness, cognitive disturbance, seizures and myoclonus. Secondly, there was no alternative cause for neurological symptoms, such as acute comorbidities or newly started medications. For CIN analysis, we included all courses of cefepime where TDM was performed.

The coadministration of neurological drugs delivered for weaning protocols was recorded on the day of dosage (dexmedetomidine, lorazepam, morphine or levomepromazine).

### 2.4. Statistical Analysis

Descriptive analysis was performed to check and summarize the data. Quantitative variables are reported as means ± standard deviations. Qualitative variables are reported as numbers and percentages. Continuous variables were compared using Student’s *t*-test. Categorical variables were compared using the chi-square test or Fischer’s exact test when the chi-square was not appropriate. Differences between groups were considered to be significant for variables yielding a *p* value ≤ 0.05. When appropriate, continuous variables were analyzed as categorical variables using clinically meaningful cut-off points (maximization of the chi-square).

A stepwise logistic regression analysis was performed to identify risk factors associated with ICU mortality. In order to identify independent risk factors for mortality, variables were included in the multivariate model if the *p* value was ≤0.05 in the bivariate analysis. Adjusted odds ratios (AORs) were computed using logistic regression analysis including the independent predictors of mortality. A receiving operating characteristic (ROC) curve was constructed to review the sensitivity and specificity of using various cefepime steady-state concentrations to predict the development of neurotoxicity. All statistical analyses were performed using SAS software, V9.4.

## 3. Results

### 3.1. Baseline Characteristics and Cefepime Treatment

In total, 152 patients were administered 157 courses of cefepime during the study period. Of the 157 courses of cefepime administered, only 78 courses had a TDM of cefepime within 24 to 48 h after starting treatment. The mean age was 67.1 ± 11.2 years. The diagnosis on ICU admission was sepsis or septic shock (*n* = 29), COVID disease (*n* = 26), acute respiratory failure (*n* = 12), neurological disease (*n* = 3) or other diagnoses (*n* = 8). Baseline characteristics and aspects of cefepime treatment are described in Table 1. The mean daily dose was 5.3 ± 1.3 g, and the dose was not adjusted for creatinine clearance in 16 patients. The median duration of cefepime therapy was 5.2 ± 2.4 days.

### 3.2. TDM of Cefepime

The mean cefepime plasma level at steady state was 59.8 ± 29.3 mg/L. The cefepime plasma level was higher than 35 mg/L in 63 (80.7%) patients. A daily dose of cefepime > 5 g and creatinine clearance < 60 mL/min were independently associated with a dose ≥ 35 mg/L (Table 2).

### 3.3. Bacterial Data

Cultures were positive in 50 patients. A total of 43 Enterobacteriaceae specimens were isolated in 33 patients, and *Pseudomonas aeruginosa* was isolated in 15 patients (Table 3). Cefepime-resistant Enterobacteriaceae was determined in four patients, and two patients presented sepsis caused by *Enterobacter cloacae* or *Escherichia coli* with a MIC of 4 mg/L. Concerning *P. aeruginosa*, cefepime resistance was determined in three patients, and a MIC of 8 mg/L was present in four patients.

### 3.4. Cefepime-Induced Neurotoxicity

The presence of CIN was assessed for 52 courses of delivered cefepime in 48 patients. Of these 52 courses, 16 (30.8%) were associated with CIN. CIN was diagnosed in the presence of altered mental status (*n* = 8), confusion (*n* = 7), reduced consciousness (*n* = 3), myoclonus (*n* = 2), coma (*n* = 2), seizures (*n* = 1) and agitation (*n* = 1) (many symptoms could be present in a patient). EEG was performed in 6 patients with disturbances in 5 patients (atypical triphasic waves (*n* = 2), severe diffuse slowing (*n* = *2*) and multifocal sharp wave (*n* = 1)). The mean time from the start of cefepime treatment to the onset of neurotoxicity was 3.3 ± 1.5 days. Neurological drugs were coadministered with 13 courses of cefepime treatment: lorazepam (*n* = 12), morphine (*n* = 10), dexmedetomedine (*n* = 5) and levomeprazine (*n* = 5). Brain imaging was performed in eight patients, with magnetic resonance imaging in three patients and CT scans in five patients. Cerebral imaging was normal in seven patients, and one patient presented preexisting subarachnoid hemorrhage. All patients returned to a satisfactory level of consciousness after a period of 2.8 ± 1.5 days with the cessation of treatment in nine patients and a reduced dose of cefepime in seven patients. Clinical and biological parameters associated with the occurrence of CIN are presented in Table 4. The mean cefepime plasma concentration was 85.7 mg/L in patients with CIN and 55.8 mg/L (*p* = 0.005) in patients without CIN. Neurotoxicity was not associated with preexisting neurological diseases.

Preexisting chronic renal failure and cefepime plasma concentration > 60 mg/mL were risk factors independently associated with the occurrence of CIN (Table 5).

ROC curve analysis indicated that a steady-state cefepime plasma concentration of 63.85 mg/L provided the best differentiation between patients who experienced CIN and those who did not (Figure 1). The lowest cefepime concentration associated with CIN was 42 mg/L.

## 4. Discussion

Targeting a free plasma beta-lactam concentration between 4 and 8 times the MIC of the causative bacteria for 100% of the dosing interval (fT ≥ 4–8 × MIC = 100%) is recommended in critical care patients [5,17]. A CI of beta-lactams is used to optimize the time during which the unbound-drug concentration exceeds the MIC, which can lead to superior clinical outcomes in some patients [18]. Although CI is not supported by strong evidence, because other studies have shown that this procedure did not improve survival or clinical cure, we have chosen it in our unit for some beta-lactams, including cefepime, when sepsis with less-susceptible microorganisms are suspected [19,20]. In a prospective multicenter observational study, Taccone et al. reported that the median percentage of T > 4 × MIC was only 34% when cefepime was delivered at a dosage of 2 g every 12 h in patients with a creatinine clearance between 50 and 80 mL/min [21]. In our study, for all patients, the plasma levels of cefepime were well above 4 times the MIC for 100% of the dosing interval, which was the pharmacodynamic target recommended in other studies to maximize bacteriological and clinical responses [22,23,24]. As in our study, a dose of 2 g every 8 h of cefepime has been used in a prospective randomized controlled trial of continuous versus intermittent beta-lactam infusion in critically ill patients with severe sepsis, and this dose has been recently recommended for the treatment of antimicrobial-resistant Gram-negative infections [10]. As the CI of cefepime needs a loading dose of 2 g, the daily dose received in the first 24 h is 8 g for patients with normal renal function. With the recent EUCAST-proposed breakpoints, targeting a cefepime plasma concentration between 30 and 40 mg/L is expected while waiting for the MIC of causative pathogens [4].

Plasma concentrations of beta-lactams are difficult to predict in ICU patients due to several physiological modifications, such as a diminution or increase in creatinine clearance, low protein binding, hypoalbuminemia or capillary permeability with plasma volume leakage. These pathophysiological disturbances result in high inter-individual variability in the cefepime volume of distribution and clearance, with the risk of achieving sub- or supratherapeutic plasma concentrations. In a prospective study assessing the TDM of cefepime in critical care patients, peak serum concentrations demonstrated a 2- to 3-fold variation, with up to a 40-fold variation in trough concentrations [25]. The mean steady-state concentration of cefepime was 59.8 ± 29.3 mg/L in our study and was largely greater than the highest established therapeutic target set at 35 mg/L [5]. Indeed, 80% of our patients had supratherapeutic levels. Few studies have described the plasma levels of cefepime in ICU patients. Many reports describe the TDM of cefepime in non-ICU patients with the measurement of trough concentrations associated with intermittent infusion [26]. Only two studies have assessed the plasma monitoring of the CI of cefepime in critically ill patients. Huwyler et al. collected cefepime TDM data in nearly thirty patients [3]. These patients were mostly non-ICU patients and received a mean daily dose of 4.1 ± 1.8 g. The mean steady-state level was 29.2 mg/L (IQR 18.9–45.9). In a more recent study, Vercheval et al. found a level of 46 mg/L with a delivered dose of 2 g every 12 h in patients with normal renal function [9]. Only half of their study population was ICU patients. The mean steady-state level of cefepime was higher in our study. The dose we used is probably too high in relation to PK/PD objectives: most isolated Gram-negative bacteria (86%) presented with a MIC to cefepime ≤ 1 mg/L. Only two patients with Gram-negative sepsis had a MIC of 4 mg/L and could benefit from plasma levels between 30 and 40 mg/L. Concerning the 15 patients with *P. aeruginosa* infections, only 4 of them with a MIC of 8 mg/L could benefit from the therapeutic range nearing 60 mg/L. These four patients accounted for 5% of our studied population.

We found that a daily delivered dose greater than 5 g/d was associated with overdosing. Other studies are necessary to determine the optimal dose of cefepime delivered by CI in ICU patients considering local bacterial epidemiology and antimicrobial resistance.

CrCL < 60 mL/min is also associated with supratherapeutic levels of cefepime. In many studies, higher cefepime plasma trough levels have been associated with impaired renal function [8,25,27]. As the elimination of cefepime is primarily mediated by glomerular filtration in the kidneys, reduced Cr/Cl has been shown to lead to drug accumulation [28,29]. Renal dysfunction increases the half-life of cefepime from 2 to 13 h, prompting the need for dose adjustments [30]. Decreasing the dose of cefepime according to Cr/Cl is recommended even if the use of algorithms may not be sufficient to prevent “toxic” concentrations of cefepime. Our results showed that supratherapeutic levels could be reached even when the dose was adjusted for renal function. In contrast, the evaluation of current dosing guidelines indicated that most simulated patients with CrCL between 11 and 29 mL/min would have subtherapeutic levels in a Monte Carlo simulation approach [31]. For these reasons, further investigations optimizing cefepime renal dosage adjustments are required.

The CIN incidence has been reported to range from 2 to 23% of patients [2,3,8,32]. As in our study, the most common risk factors include renal failure and an overdose of cefepime, although neurotoxicity has been described in patients with normal renal function or with a dose adjusted for CrCl [32,33,34]. In our study, the CIN incidence was higher, reaching 30% of patients. This high rate is partly explained by the large number of patients with supratherapeutic levels of cefepime. Moreover, we studied relatively old patients with a high proportion of renal impairment (54%) and included only patients without deep sedation. Likewise, we cannot exclude encephalopathy mediated by the inflammatory response to sepsis or by drugs used for the weaning protocol, such as morphine or neuroleptics, although there was no difference between the two study groups.

There is considerable variability in the cut-off serum cefepime level to predict the incidence of CIN. The trough levels of cefepime associated with CIN range between 7 and 36 mg/L and depend on the delivered dose, renal function and the severity of the illness [6,7,8]. We found a higher cefepime toxicity threshold. A cefepime plasma concentration ≥ 60 mg/L was associated with the occurrence of CIN. The ROC curve analysis confirms this level with a cut-off point of 63.8 mg/L. Our result is similar to the results of Vercheval et al., where a cefepime steady-state concentration of 63.2 mg/L was the best cut-off point between patients with and without neurotoxicity when cefepime was delivered by CI [9]. In the study by Payne et al., 26% of patients experienced neurotoxicity despite appropriate dosing [2]. In our study, the lowest cefepime threshold without neurotoxicity was 42 mg/L. This result matches with other studies, where no toxicity occurred below trough concentrations of 35 mg/L and 38.1 mg/L [6,8].

The median time to CIN diagnosis is generally 4 days with a progressive course [2]. In our study, the median duration from drug initiation to CIN development was 3.3 ± 1.5 days. As in other studies, the prognosis of CIN was favorable. Most cases usually improve with a median time after the cessation of cefepime of 3 days [26]. In our patients, improvement in consciousness was effective after 2.8 ± 1.5 days. We found no excess mortality, unlike other studies, where CIN was associated with a mortality ranging from 13% to 43% [2,35,36]. Another discordance with the literature concerns the role of underlying cerebral diseases as a predisposing factor for CIN. We found no association, whereas patients with underlying brain diseases had a four-fold increased risk of neurotoxicity in the study by Vercheval et al. [9].

Our study has several limitations requiring some comments. First, it is a retrospective study including a limited number of patients. Our study probably lacks power, and other risk factors could have been highlighted with a larger number of patients. Similarly, our data were not specifically collected to determine the incidence of CIN, and although we have used a strict definition, we cannot be sure of the exclusivity of cefepime in the occurrence of neurological disorders. Alternative causes of adverse neurologic events are possible, such as renal impairment, sepsis-associated encephalopathy or the concomitant use of other antimicrobials. Second, this study was performed in one ICU, and the findings of this report might not be representative in other settings. Third, beta-lactam TDM was implemented in our unit in 2019 and was not optimal despite its encouragement to be performed. A limitation is that 79 courses were excluded because of absent plasma sampling. Fourth, there is no accepted definition of a therapeutic cefepime steady-state concentration, and the level we have chosen (35 mg/L) can be discussed depending on the main isolated pathogens and their sensitivity. Moreover, the exact steady-state concentration at the time of suspected CIN was not possible for all patients; hence, we selected the closest TDM result at the time of toxicity presentation. Fifth, renal function may be overestimated by the CKD-EPI equation, particularly in critically ill patients.

## 5. Conclusions

Overdosing is frequently observed in critically ill patients receiving high doses of cefepime and is associated with renal impairment and a daily dose >5 g. Renal failure and a cefepime plasma concentration ≥ 60 mg/L are associated with the occurrence of CIN.

## Figures and Tables

**Figure 1 antibiotics-12-00069-f001:**
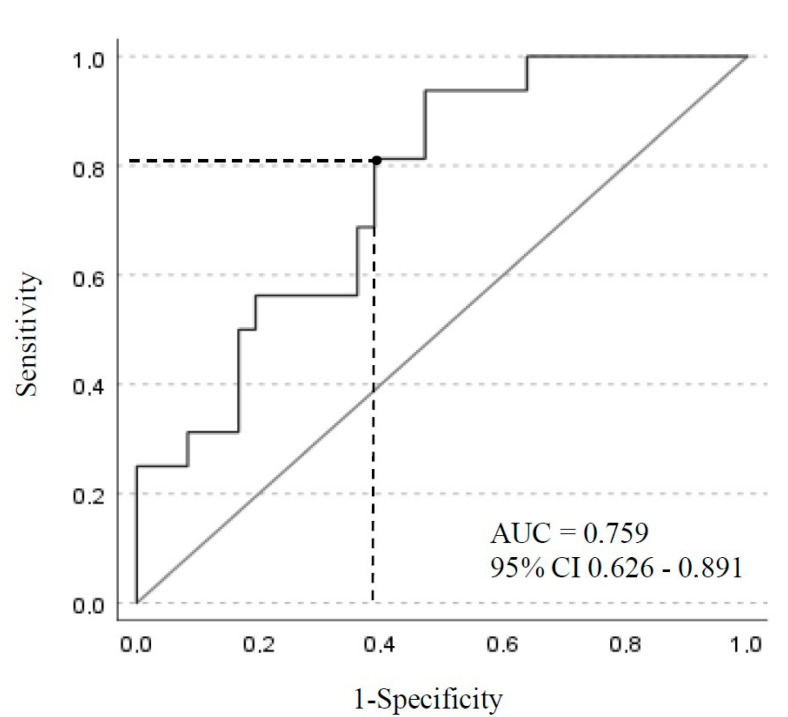
Receiving operating characteristic curve of cefepime plasma concentration with respect to neurotoxicity. AUC, area under the curve; 95% CI, 95% confidence interval.

**Table 1 antibiotics-12-00069-t001:** Clinical characteristics of patients and predisposing factors associated with overdosing of cefepime.

Variable	Dosage < 35 mg/mL *n* = 15	Dosage ≥ 35 mg/mL *n* = 63	*p* Value
Age (years)	63.3 ± 10.4	68.0 ± 11.2	0.06
Age > 65 years	5 (33.3)	42 (66.6)	0.01
Male sex	11 (73.3)	47 (74.6)	0.91
Charlson score	3.2 ± 1.6	4.0 ± 2.5	0.41
SAPS II	31.9 ± 7.8	42.4 ± 15.2	0.10
BMI (kg/m^2^)	28.3 ± 7.2	31.2 ± 8.9	0.25
Chronic renal failure	1 (6.6)	8 (12.7)	0.51
Underlying neurological disease	2 (13.3)	11 (17.4)	0.69
Cirrhosis	0 (0)	4 (6.3)	0.31
Diabetes	4 (26.6)	19 (30.1)	0.78
Cardiac chronic failure	4 (26.6)	20 (31.7)	0.70
Indication for cefepime treatment			
Nosocomial pneumonia	8 (53.3)	42 (66.6)	0.33
IAI	2 (13.3)	6 (9.5)	0.66
Bone and joint infection	4 (26.6)	10 (15.9)	0.32
Other	1 (6.6)	5 (7.9)	0.86
Start of cefepime treatment			
Duration of ICU stay before treatment (days)	11.0 ± 14.0	13.7 ± 13.8	0.27
Serum albumin (mg/L)	23.0 ± 6.2	21.6 ± 6.2	0.53
SOFA score	4.8 ± 2.5	4.9 ± 2.3	0.85
Creatinine clearance (mL/min)	90.9 ± 34.2	67.8 ± 39.7	0.02
Creatinine clearance ≤ 60 mL/min	3 (20)	31 (49.2)	0.04
Daily dose (g)	4.6 ± 1.2	5.4 ± 1.2	0.005
Daily dose > 5 g	6 (40)	52 (82,5)	0.0007
Dose adjusted for creatinine clearance	14 (93.3)	48 (76.2)	0.13
Duration of ICU stay (days)	25.4 ± 15.7	33.6 ± 30.8	0.56
Number of deaths	2 (13.3)	12 (19)	0.5

Data are presented as *n* (%) or mean ± SD; SAPS: simplified acute physiologic score; BMI: body mass index; IAI: intra-abdominal infection; SOFA: sepsis-related organ failure.

**Table 2 antibiotics-12-00069-t002:** Logistic regression analysis of independent risk factors associated with overdosing of cefepime.

Variable	Odds Ratio	95% CI	*p* Value
Creatinine clearance < 60 mL/min	8.0	1.52–42.6	0.01
Daily dose of cefepime > 5 g	11.2	2.44–51.28	0.001

**Table 3 antibiotics-12-00069-t003:** Microorganisms isolated in patients with microbiologically proven infections with their cefepime MICs according to EUCAST clinical breakpoint defining susceptibility or resistance.

Microorganism	MIC ≤ 0.001 mg/L	MIC ≤ 1 mg/L	MIC = 4 mg/L	MIC = 8 mg/L	MIC > 8 mg/L
*Proteus species*	.	5	.	.	.
*Enterobacter species*	.	13	1	.	2
*Klebsiella species*	.	3	.	.	2
*Escherichia coli*	.	6	1	.	.
*Serratia species*	.	3	.	.	.
*Morganella morganii*	.	2	.	.	.
*Citrobacter species*	.	3	.	.	.
*Hafnia alvei*	.	2	.	.	.
*Pseudomonas aeruginosa*	8	.	.	4	3

Data are presented as the number of isolated pathogens. Enterobacteriaceae was susceptible to MIC ≤ 1 mg/L and resistant to MIC > 4 mg/L, and *Pseudomonas* spp. were susceptible to MIC ≤ 0.001 mg/L and resistant to MIC > 8 mg/L (ref EUCAST).

**Table 4 antibiotics-12-00069-t004:** Univariate analysis defining clinical and biological characteristics associated with neurotoxicity in 52 courses of cefepime treatment.

Variable	Courses with Neurotoxicity (*n* = 16)	Courses without Neurotoxicity (*n* = 36)	*p* Value
Male sex	10 (62.5)	26 (72.2)	0.48
Mean age (years)	72.3 ± 10.2	68.0 ± 10.6	0.21
Charlson score	6.0 ± 2.8	3.7 ± 1.9	0.01
SAPS II	49.3 ± 14.6	39.1 ± 14.3	0.04
BMI, kg/m^2^	31.6 ± 5.6	30.9 ± 9.2	0.63
Chronic renal failure	6 (37)	3 (8.3)	0.01
Cirrhosis	3 (18.7)	1 (2.7)	0.04
Diabetes mellitus	8 (50)	10 (27.7)	0.12
Chronic cardiac failure	5 (31.2)	15 (41.6)	0.47
Previous brain disease	3 (18.7)	5 (13.9)	0.65
ICU admission			
Covid disease	1 (6.2)	7 (19.4)	0.22
Acute respiratory failure	2 (12.5)	9 (25)	0.31
Sepsis	8 (50)	17 (47.2)	0.85
Neurological disease	2 (12.5)	0	0.03
other	3 (18.7)	3 (8.3)	0.27
Concomitant use of neurosedative drugs	4 (25)	9 (25)	1
Dexmedetomedine	1 (6.2)	4 (11.1)	0.58
Morphine	2 (12.5)	8 (22.2)	0.41
Benzodiazepine	2 (12.5)	10 (27.7)	0.22
Neuroleptics antipsychotics agents	0	5 (13.9)	0.11
Daily dose of cefepime (g)	5.1 ± 1.4	4.9 ± 1.4	0.75
Creatinine clearance (mL/min)	43.7 ± 33.1	72.1 ± 37.1	0.02
Creatinine clearance < 60 mL/min	13 (81.2)	16 (44.4)	0.01
Dose adjusted for renal function	10 (62.5)	28 (77.7)	0.25
Presumed infection			
Lower respiratory infection	9 (56.2)	18 (50)	0.67
IAI	1 (6.2)	4 (11.1)	0.58
Osteoarthritis	4 (25)	10 (27.7)	0.83
Other/undetermined	2 (12.5)	4 (11.1)	0.88
SOFA score on the day of dosage	4.8 ± 2.2	4.5 ± 2.6	0.76
Delivery of cefepime (days)	3.3 ± 1.5	3.0 ± 2.4	0.11
Cefepime plasma concentration (mg/L)	85.7 ± 32.4	55.8 ± 24.9	0.005
Duration of ICU stay (days)	26.3 ± 33.1	25.6 ± 29.3	0.72
Number of deaths	1 (6.2)	3 (8.3)	0.79

Data are presented as *n* (%) or mean ± SD; SAPS: simplified acute physiologic score; BMI: body mass index; IAI: intra-abdominal infection; SOFA: sepsis-related organ failure.

**Table 5 antibiotics-12-00069-t005:** Logistic regression analysis of independent risk factors associated with occurrence of cefepime-induced neurotoxicity.

Variable	Odds Ratio	95% CI	*p*
Chronic renal failure	7.0	1.27–38.6	0.02
Plasma concentration ≥ 60 mg/mL	5.6	1.24–26.1	0.02

## Data Availability

The data presented in this study are available upon reasonable request from the corresponding authors.

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
