# Peer review of "Continuous Infusion of High Doses of Cefepime in Intensive Care Unit: Assessment of Steady-State Plasma Level and Incidence on Neurotoxicity"

_antibiotics, 2022, doi:10.3390/antibiotics12010069_

Round 1
Reviewer 1 Report
Reviewer report
Manuscript title: “Continuous infusion of high doses of cefepime in intensive care unit: assessment of steady -state plasma level and incidence on neurotoxicity .”
Thank you for the opportunity to review the manuscript ID :antibiotics- 2074202
Vanessa Jean-Michael and collaborators reported a retrospective original study included 78 patients treated Cefepime according to TDM ( 79 other courses have been excluded ). The manuscript is informative since it presents still very important issue in the area of antibiotic PK/PD at ICU patients. Despite the numerous articles on continuous infusion of beta lactams antibiotics had been published, it is still very rare to find published PK/PD studies in the last years about Cefepime in CI. The most original aspect of this manuscript is the assessment of steady state plasma level of Cefepime in CI and additionally very difficult to diagnosed incidence of neurotoxicity at ICU patients.
In general, the study is very interesting well performed, the question posed by the authors is well defined and the data is sound. The title and the abstract accurately convey what has been found. Additionally, the results in my opinion, are quite well described and presented in 5 tables. The several limitations of the study are clearly stated, and sufficiently discussed in discussion section. In my opinion the authors should heavily underline that at severe ill patients several factors could be responsible for neurological disorders as well as because it was retrospective study selection bias was possible . Discussion is interesting. The manuscript is supported by 34 references, and 10 of them had been published between 2018-2022 . Conclusions are adequately supported by the analyzed data .
The manuscript adheres to relevant standards for reporting and data deposition.
Minor comments :
Material and Methods section should be usually located after Introduction – check please guideline for “Antibiotics “
Please unify SI units “ g or gr “ in text and tables -page1,2,3,6,8
Please correct in Table 1 – Creatinine clearance should be in mL/min ( not /mn)
Overall, with a pleasure I can recommend this study for publication after this small correction.
Reviewer 2 Report
Thank you for giving us an opportunity to review your research. Please see comment below:
1. Encourage to proof read the manuscript for minor grammatical errors and use standard abbreviations e.g. for minute use min. or give a reference to abbreviations
2. Sections of results. Outcomes 2.5 : Are these the final outcomes ( mentioning length of stay) of the studies along with the aim mentioned in introduction? If they are the aim of the study, it should be mentioned in the Aim section along with aim 1 and aim 2. If they are secondary variables being evaluated in the study, should also be mentioned in the introduction/Aim
3. Discussion sections talked about higher rate of neurotoxicity in the research but does not mention the finding that it could also be secondary to supratherapeutic levels of CEfepime seen in 80 % of patients
4. PLease review the conclusion. Encouraged to modify conclusion per questions asked in the study
